# Revisiting the physical limits to economic growth, with a focus on the waste heat limit

**Dorian S. Abbot**[1]⊚*, **Anup Malani**[2,3]⊚

**1** Department of the Geophysical Sciences, The University of Chicago, Chicago, Illinois, United States of America, **2** Law School, The University of Chicago, Chicago, Illinois, United States of America, **3** National Bureau of Economic Research, Cambridge, Massachusetts, United States of America

⊚ These authors contributed equally to this work.
\* abbot@uchicago.edu

**Data availability statement:** The data are publicly available and we have placed the code to analyze them on GitHub with a link in the paper. Our data and code are available at https://github.com/anup-malani/limits-to-growth/.

**Funding:** The author(s) received no specific funding for this work.

## Abstract

Whether there are imminent physical limits to economic growth has been the focus of a long-standing debate, often (but not exclusively) pitting natural scientists against economists. One side of the debate posits that finite resources constrain human economic growth, the other that productivity improvements and substitution can relax those resource constraints. However, the two sides lack a common framework to resolve their disagreement. We provide this framework and mediate this dispute using a combination of low-order economic and physical models, focusing especially on the waste heat limit. To start, we use a simple accounting identity to show that historical rates of growth in average productivity can support modest economic and population growth (totaling 1.2%). We model the future trajectory of productivity growth by combining a canonical production function (Cobb-Douglas) with a flexible model where population growth drives total-factor-productivity growth. We find remarkable parameter sensitivity: across a plausible range of parameters waste heat either never constrains economic growth in the model or does so in a few hundred years. Interestingly, either decreasing or increasing population growth can dramatically extend the time to heat constraint in the model, the latter due to increases in innovation. Finally, we use a more flexible (constant elasticity of substitution) production function and historical changes in the relative use of different resources to show that humans have the ability to substitute away from resources in shorter supply. Although there are vast uncertainties in this type of projection, our work highlights the potential for an optimistic future for humanity in which economic growth continues on a millennial timescale.

## Introduction

Are there physical limits to economic growth, defined as increases in per capita income or output? This question has been the focus of a long-standing debate, often (but not exclusively) pitting natural scientists against economists. Following Malthus [1], the affirmative

**Competing interests:** The authors have declared that no competing interests exist.

position is that limits on Earth's resources place an imminent cap of human potential for sustained growth in consumption. The negative position is that due to continued productivity improvements and resource substitution, physical limits may not bind economic growth for the foreseeable future. Concerns about climate change and declines in human fertility rates have renewed the debate, which remains unresolved.

Famously, the 1972 report *The Limits to Growth* [2] contained simulations showing that, if population growth and resource consumption continued at the then-current rates, the human population could not grow for more than 100 years without exhausting available resources. The simple intuition was that human population and per capita resource utilization were rising at exponential rates, while total resources were finite. Growth would cease when total resource use exhausted available resources. That report built on prior work in the post-war period [3] that, along with popular texts such as Ehrlich and Ehrlich's *The Population Bomb* [4], presented scenarios predicting famines and other mass-casualty events before the turn of the millennium as human populations hit resource limits. Murphy [5] has updated the argument in *The Limits to Growth* to account for more recent data and specific resource constraints. Specifically, he argued that economic growth is tied to power consumption and is therefore limited: if power consumption continues to grow exponentially at its current rate for the next ∼400 years and the associated waste heat is dumped on Earth's surface, Earth will become too hot for life.

Critics contend that *The Limits to Growth* ignored improvements in productivity [6]. These can be measured by, for example, the prices of resources or commodities. Moreover, the economic growth literature suggests that population size itself, in particular the number of highly innovative individuals, is partially responsible for growth in productivity, and thus per capita income [7–9]. Finally, recent evidence suggests that even previously fast-growing populations in low-income countries are completing the demographic transition (decrease in total fertility rate), leading to stabilization and likely future decline of the global population [10–12]. All of these points suggest that economic growth may be possible for an extended period.

An unstated assumption of literature such as *The Limits to Growth* is that humanity will remain limited to Earth. However, since the mid-20th century, there has been extensive speculation about the ability of human civilization to spread throughout the Solar System and beyond [13–16], exploiting additional resources and energy as it goes. This conversation has been enhanced by the discovery of increasingly Earth-like extrasolar planets this century [17]. In fact, even Murphy's [5] limit of consuming all the power generated by the Milky Way galaxy is simply the definition of a Type III Civilization on the Kardashev Scale [14]. Moreover, Murphy's [5] waste heat argument would clearly be moot for a civilization that had spread beyond Earth, because the waste heat would not be deposited on Earth's surface. Or we may even be able to avoid the waste heat problem simply by transferring power creation or industry to space without moving the human population. For the purposes of this paper, we will bracket this issue and proceed to examine the problem from the perspective of an Earth-bound civilization, but it is important to bear in mind this caveat.

In this paper, we reexamine a critical component of the physical limit to growth: the power usage, defined as usage of energy per unit of time. First, we present a simple accounting identity that relates total power consumption to human population, per capita income, and average efficiency of power exploitation (defined as power used to generate a dollar of income). We use this identity to place bounds on economic and population growth given historical rates of growth in the average productivity of power. Next, we consider a more sophisticated model that includes variation in future productivity to examine how sensitive estimates of when humans will hit the waste heat limit are to parameters that describe the relative importance of different inputs (including power) into production, the population growth rate, the

rate at which people generate innovations, and how much harder incremental innovation is becoming. We find large sensitivity to these parameters, with plausible scenarios in which economic growth continues without ever encountering the waste-heat limit. We interpret such infinite times to the waste-heat limit in these simple models as indicating scenarios where the waste-heat limit does not constrain economic growth for at least a thousand years, which is sufficient to diffuse the immediacy of calls for limits to growth. Finally, we examine the potential for humans to substitute away from power – and other limited resources – and continue economic growth despite specific resource exhaustion. In our discussion, we examine limits to growth other than waste heat and implications of our analysis for policy. There may be other limits to growth, but the discussion of when those limits will bind growth will be qualitatively similar to the discussion of limits due to waste heat. Ultimately, we conclude that limits certainly exist, but more research is required to determine when they will bind.

Because this paper attempts to speak across academic disciplines, it is useful to define some technical terms. Gross domestic product (GDP) or output is a non-physical quantity equal to the physical output of goods and services times the market prices of those goods and services in a period of time and in a location. For example, consider the hypothetical country Simplexia which produces only two outputs, a physical good $q_1$ (housing, measured in square meters) and a service $q_2$ (entertainment, measured in hours). Each of these outputs is produced using a physical input $z$ (land, measured in $m^2$), so $q_1 = a_1 z$ and $q_2 = a_2 z$, where $a_1$ and $a_2$ describe the number of units of each output produced from 1 $m^2$ of land. Each output is associated with prices $p_1$ (measured in dollars per $m^2$) and $p_2$ measured in (dollars per hour). The value of Simplexia's GDP is $p_1 q_1 + p_2 q_2$, measured in dollars. The annual national or global output extrapolates this logic to all goods and services produced by a real country or all countries in a given year. (The supplemental information (SI) explains what a dollar is, how it is valued, and what economists mean when they speak of real or purchasing power parity adjusted dollars. The SI also explains why economists use output as a proxy for income.) Per capita GDP is defined as GDP divided by population. Economic growth is growth in GDP or GDP per capita per unit of time.

Given this definition of economic growth, there are at least two ways to intuitively understand how economic growth (increase in dollar value GDP) is possible with finite physical resources (e.g., constant amount of land, $z$, in Simplexia) by turning input into output more efficiently. First, improvement in productivity reduces the amount of finite physical input required to produce a good or service. For example, engineering advances could allow the building of multi-story structures (an increase in $a_1$), increasing the quantity of housing ($q_1$) produced per $m^2$ of land ($z$). Alternatively, the invention of television means that a play on one stage or a basketball game on one court can entertain many more people (an increase in $a_2$), increasing the amount of entertainment ($q_2$) that can be produced from each $m^2$ of land ($z$). Second, consumers may change their consumption away from a product that uses more physical resources per unit to one that uses fewer physical resources per unit. For example, suppose $a_1 < a_2$, so that producing housing is more land-intensive than entertainment. Holding output prices constant, if one $m^2$ of land is shifted from production of housing to production of entertainment, housing quantity falls by $a_1$ but entertainment quantity rises by $a_2$ hours. If $p_1/p_2 < a_2/a_1$, i.e., citizens of Simplexia value entertainment enough relative to land, then this would increase Simplexia's GDP.

Before we present our analysis, a word of caution is necessary with respect to long-term growth projections and predictions concerning the decline and fall of human civilization, including those here. The economy, including demographics, preferences, environment, and technology are complicated. Yet here we use fairly simplistic predictive formulas to describe them. Nor are complicated models always superior. The complex simulation that motivated

*The Limits to Growth* report [2] was bested by simple analytical models [6]. Moreover, we have historically not been able to predict many phenomena that ended up having first-order effects on global welfare. Malthus wrote his famous "An Essay on the Principle of Population" [1] just as his eponymous era was displaced by the Industrial Revolution [18]. Decades after the demographic transition occurred in England, but before it brought China and India's fertility to near replacement levels [11], Ehrlich projected continued exponential population growth [4]. And, just 25 years ago, global warming was a novel concept.

Given this caution, one might question the utility of this paper. However, our analysis is consistent with our warning and provides a road map for how to reduce error in projections. First, this paper tests the robustness of claims about the limits to economic growth, which are often made with great confidence [2,4]. Second, this paper provides specific qualitative guidance on how more realistic dynamics of population growth, productivity growth, and technological adaptation affect the time to resource exhaustion or waste-heat limitation. In summary, *the Limits to Growth* literature argues for a short future of economic growth with a small uncertainty. Our work suggests that a much longer future of economic growth is appropriate, but also that the uncertainty is large.

## Materials and results

### Potential for economic and population growth without power growth

Suppose humans stopped increasing power consumption. How much would this limit economic or population growth?

Total energy consumption by humans per unit of time, i.e., aggregate power consumption ($P$), is identical to the product of: the human population ($N$), per capita output (income) per year ($y$), and the average amount of power used to produce a unit of output, which is equivalent to the reciprocal of average energy efficiency or productivity, $(\overline{A_p})$ measured in economic output per unit of energy,

$$P = \frac{N \cdot y}{\overline{A_p}}.$$ (1)

The units in Eq (1) are W = (people) x (USD/person) x (W/USD), where W stands for Watts ($kg\ m^2\ s^{-3}$), or Joules per second, and USD stands for real, purchasing power parity-adjusted United States Dollars. Taking the logarithm and time derivative, we can rewrite Eq (1) in terms of growth rates:

$$g_P = g_N + g_y - g_{\overline{A_p}},$$ (2)

where the growth rate of quantity $x$ is written $g_x \equiv (\partial x/\partial t)/x$. $g_y$, the growth in per capita income, is the main measure of economic growth.

We estimate the geometric mean of power, population, and per capita income growth rates in this equation for the period 1965 to 2020 using global data on power from Our World in Data [19] and on population and per capita income from [20] in Fig 1. We calculate the implied growth rate of average productivity using Eq (2). Power consumption is increasing at a rate of 2.07%/yr, slightly lower than Murphy's [5] estimate of 2.3%/yr. Murphy's slightly higher estimate results from adding data from the first half of the 1900s from [21] to our data. Either estimate is nearly enough to increase power consumption by a factor of 10 every century. Interestingly, the growth in power consumption results from 1.73% economic growth and 1.54% population growth per year, offset by 1.2% growth in average productivity per year.

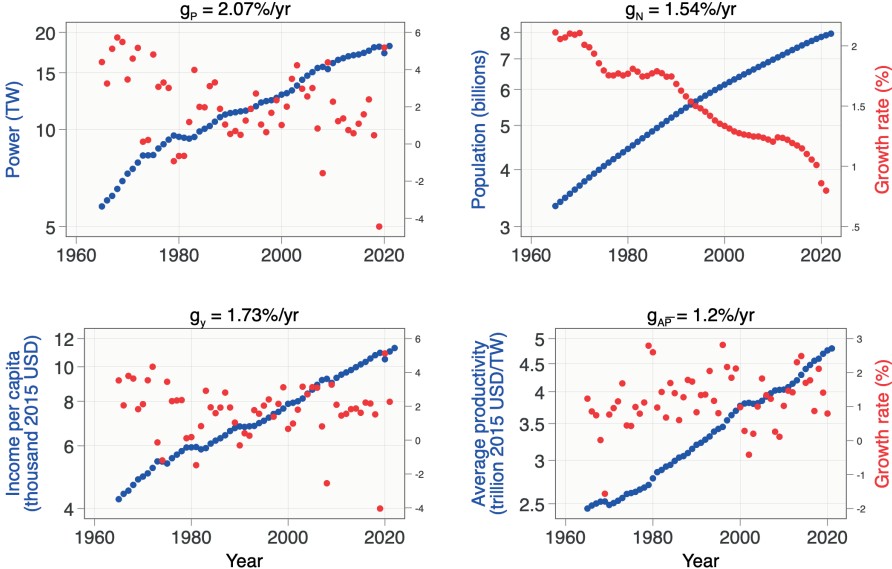

**Fig 1. Global power consumption (*P*), population (*N*), per capita output (*y*), and average productivity of power ($A_P = (y \cdot N)/P$) from 1965–2020.** Plot titles give geometric mean of growth rate in power ($g_P$), population ($g_N$), per capita output ($g_y$), and average productivity of power ($g_{\overline{A_P}}$). Power data are from Our World in Data and population and output per capita data are from the World Bank.

It is interesting to consider the possibility of population and economic growth in the limit of zero growth in power consumption. Any combination of $g_N$ and $g_y$ that yields $g_N + g_y = g_{\overline{A_P}}$ will result in $g_P = 0$. Let us consider a scenario in which the recent historic growth rate in average productivity simply continues. If the population stays constant, per capita income could continue to grow at 1.2%/yr, roughly 69% of historical growth rates, without increasing energy consumption simply by exploiting increases in energy efficiency. Alternatively, if per capita income stopped increasing, the population could grow by 1.2%/yr, roughly 78% of its current growth rate, again only by exploiting increases in energy efficiency without increasing energy usage. Evidently, if historical increases in efficiency continue, they will be capable of driving significant economic and population growth even in the absence of growth in power consumption. Of course one cannot "choose" future growth in power usage, economic growth, and population growth. But this analysis suggests it is physically possible to have reasonable population and economic growth—even if power growth is contained–if historical progress in efficiency continues.

## Delaying the limits to economic growth with slowing population growth

Those positing physical limits to growth often forecast dates by which human civilization will exhaust a key resource [4,5]. These forecasts tend to be based on forecasts of population growth from a period where death and birth rates diverged, so the population grew rapidly. However, the demographic transition has brought birth rates back down to death rates, or even below, in many countries [22].

The median forecasts of the United Nations, the Institute for Health Metrics and Evaluation, and the International Institute for Applied Systems Analysis all agree that global population growth will reach zero then become negative on a decadal timescale [11,12,22]. Although

there is significant uncertainty in this type of forecast, particularly in the effect of high fertility sub-populations [23], these projections do motivate the consideration of a zero population growth scenario.

How much would zero population growth delay the waste heat limit to power growth? Murphy [5] estimates the time to waste-heat constraint based on the surface temperature of Earth reaching the boiling point of water (373 K) due to waste heat from power usage. Additional heating by increased greenhouse gases can be ignored because power usage (and waste heat required to generate and use that power) is assumed to grow exponentially so that it eventually dwarfs other terms. Solar geoengineering would not be a solution because, even if all sunlight were blocked, the time to the temperature limit would only marginally lengthen due to the exponential growth in waste heat. Per Murphy [5], the surface temperature $T$ (in Kelvin) of the Earth in $t$ years can be approximated by:

$$T = \left( \frac{F_\odot(1 - a) + P_0 e^{g_P(t - t_0)}}{4\sigma} \right)^{1/4} + \Delta T_{GHG}, \tag{3}$$

where $F_\odot = 1,360$ W m$^{-2}$ is the solar flux at the top of the atmosphere, $a \approx 0.293$ is the planet's albedo (reflectivity), $P_0 \approx 0.1$ W m$^{-2}$ is 4 times the power output per unit area of Earth's surface at time $t_0 = 2000$, $\sigma \approx 5.67 \times 10^{-8}$ W m$^{-2}$ K$^{-4}$ is the Stefan-Boltzmann constant, and $\Delta T_{GHG} \approx 33$ K is an adjustment that accounts for greenhouse gas warming. Murphy assumes that future growth in power will match historical growth in power, which is $g_P \approx 2.3\%$/yr in the data he examines, leading to the projection that Earth will reach the waste heat limit to growth in $\sim 400$ years.

Murphy [5] acknowledges that Eq (3) is an extreme simplification of the climate system intended only for a qualitative estimate. It is worth considering a more sophisticated global climate model that includes three-dimensional variation, moisture, detailed radiative physics, and clouds. The physical process that would certainly limit human habitability is the runaway greenhouse [24], which is associated with the surface temperature reaching 373 K, but not exactly the same thing. Different global climate models suggest that $F_\odot$ must be increased by between 10% [25] and more than 21% [26] in order to cause a runaway greenhouse, most likely due to differences in cloud behavior. If we make the approximation that waste heat from power generation affects the climate equivalently to solar heating, following [5], we arrive at a time until the runaway greenhouse ($\tilde{t}$) via consideration of planetary energy balance as

$$\tilde{t} - t_0 = \frac{1}{g_P} \log\left( \frac{f(1 - a)F_\odot}{P_0} \right), \tag{4}$$

where $f$ is the fractional increase in $F_\odot$ necessary to cause a runaway greenhouse without excess heat from power generation. Using $g_P = 2.3\%$/yr, we obtain a limit to growth due to the runaway greenhouse in $\sim 300$ years whether we assume $f = 0.1$ or $f = 0.21$, confirming that Murphy's methodology [5] yields the correct order of magnitude.

Assuming that $g_N = 0$ and that $g_y$ and $g_{\overline{A_P}}$ remain at their recent historical values (1965–2020) yields $g_P = 0.53\%$/yr. At this lower growth rate, it will take $\sim 5$ times longer to reach uninhabitable conditions, i.e., between $\sim 1500 - 2000$ years, providing an additional millennium of runway. Alternatively if the economic growth rate were reduced from $g_y = 1.73\%$/yr to $g_y = 1.36\%$/yr, a scenario we discussed in the last section, only a 27% reduction, the time to reach uninhabitable conditions due to power consumption would be delayed indefinitely in the model.

If population growth actually turns negative, the deadline can be completely lifted in a different way. In the "Empty Planet" solutions of Jones [27], human population can get stuck on a pathway of exponential collapse while GDP per capita increases and resource use is constant. If $g_N < 0$, then per capita income can grow indefinitely in the model with zero increase in power usage at the rate of $g_{\overline{A_P}}$ – $g_N$. If current rates of productivity improvement continue, that means we can sustain existing rates of economic growth without increases in power consumption if the population growth rate is –0.53%/yr. Of course it should be noted that global population collapse is not the optimal solution to the problem of unbounded increases in power consumption from a human perspective.

Long-term projections of climate, economic growth, and population are notoriously unreliable, so one should not take these deadlines as specific quantitative predictions. Instead, the appropriate take-away from this analysis is that plausible reductions in population growth can have multiplicative effects on the waste heat deadline, possibly even extending it to the distant future, especially when coupled with slight reductions in the economic growth rate.

## Future productivity and limits to growth

A concern with relying on recent historical improvements in energy efficiency to sustain per capita income growth is that productivity growth might be slowing. The deceleration may be due to the exhaustion of low-hanging fruit or the bureaucratization of industry and regulatory approvals [28]. Has the literature found deceleration in productivity growth? How does it vary across sectors? How do we project that into future productivity growth? And how does this affect estimates of the time limits to growth?

To address these questions, we first elaborate on how power is transformed into output. Up to now we have considered a simple model in which economic output is related only to power consumption (Eq 1). Economists, however, posit that output is an increasing function of number of workers or labor ($L$), $M$ different capital inputs ($\{K_i\}$) such as power ($K_P$), and *total factor* productivity ($A$): $Y = F(A, L, \{K_i\})$. Total factor productivity (TFP) is not just productivity of a single input such as power, but productivity of all inputs combined. TFP represents things like knowledge or technology, organization of firms, markets, and regulation, which all affect how well inputs can be translated into output. Our production function uses labor or employed population ($L$) rather than population ($N$), which was used in the identity in (1), because only employed persons contribute output.

Common economic growth models often further assume that, in the long run, outputs have Cobb-Douglas form [29], i.e., are log-linear in inputs

$$Y = A \cdot \Pi_{i=0}^{M} K_i^{\theta_i} \cdot L^{\theta_L} \tag{5}$$

where $\theta_i \in (0, 1) \; \forall \; i$ and $\Theta = \Sigma_{i=0}^{M} \theta_i + \theta_L$. Here, $Y$ is measured in dollars, each capital input $i$ is in its unit $U_i$, and total factor productivity (TFP), A, is in USD/(person$^{\theta_L} \cdot \Pi_i U^{\theta_i}$). The reason for this functional form assumption is that the share of output spent on capital and labor have been remarkably stable over the last century [18] despite growth in TFP, and profit maximization or cost minimization with Cobb-Douglas production using stable $\{\theta_i\}$ but time-varying $A$ is consistent with that observation. While there are criticisms of the Cobb-Douglas function [30,31], we use it mainly to illustrate a challenge with mapping productivity changes to time limits on growth; the same qualitative behavior can be achieved with more general production functions, albeit with more mathematical complexity that distracts from our point here. The ($\{\theta_i\}, \theta_L$) are often called capital and labor shares because profit maximization or cost minimization with Cobb-Douglas technology implies that spending on each input is a

fixed share of output $y$ corresponding to the exponent on that input. We assume $\Theta = 1$, which implies constant returns to scale in inputs, because that is also consistent with long-term data [32]. If we take logs and then time derivatives of (5), we obtain an equation that relates the growth of output to growth in TFP and inputs:

$$g_{y_L} = g_Y - g_L = g_A + \theta_K(g_K - g_L) + \theta_P(g_P - g_L). \tag{6}$$

Here, $g_{y_L}$ is growth in output per employee. For simplicity, we have aggregated the stock of all capital inputs other than power and called it non-power capital: $\sum_{i \neq P} K_i = K$. We define $\theta_K = \Sigma_{i \neq P}\theta_i$ as non-power capital share. We call power $K_P = P$ and power factor share $\theta_P$.

We can estimate historical factor shares ($\theta_K, \theta_P$) and TFP growth ($g_A$) consistent with our assumption of a Cobb-Douglas production function using panel data (with annual data on a fixed number of countries) on inputs and outputs from multiple countries across prior years. Specifically, we estimate (6) using ordinary least squares linear regression and data on power consumption from the U.S. Energy Information Administration [19] and on output, employed populations, capital stock, and labor share from 66 countries consistently in the Penn World Tables (PWT) 10.01 [33] from 1965–2019. From 1965–1991, this includes the USSR. From 1965–1991, we substitute Russia plus the former Commonwealth of Independent States for the USSR, for a total of 79 states. These countries account for roughly 94.4% of global output, 80.2% of total population and 96.3% power consumption in 2019. Although TFP growth is not observed directly in our data, it can be approximated by the estimated constant in our regression. We estimate that power's share of output ($\theta_P$) is 1.99%, the remaining capital share ($\theta_K$) is 53.81%, and the labor share ($\theta_L$) is 44.2% (see S1 Table). Our estimate of power share is close to a recent measurement of 4% by Thunder Energy [34]. Our labor share estimate is somewhat low (thus our capital share estimate somewhat high) relative to the literature ($\approx 2/3$) [29], but the labor share has been declining and the capital share rising since the 1980s such that [35] estimates labor share $\leq 0.6$ in the 4 largest economies. Importantly, our estimates imply that historic TFP growth was 1% during this period, similar, but slightly smaller than our estimate of 1.2% growth in average productivity of power globally.

We are interested in future—not past—TFP growth. This can be forecasted from historical data by modeling the drivers of innovation. Many economic growth models [27,36] posit that innovation is related to the amount of working people and existing ideas:

$$g_A(t) = \frac{\dot{A}(t)}{A(t)} = \alpha L(t)^\lambda A(t)^{-\beta} \tag{7}$$

where $\dot{X} = \partial X/\partial t$, $t$ is time, and $\alpha > 0$. This equation uses working population rather than total population because the non-working population is thought not to contribute to production and thus to improvements in total factor productivity. In what follows, we allow TFP growth to vary over time, but we continue to hold working population, income and non-power capital stock growth constant. (7) permits the returns to working population to be diminishing if $\lambda < 1$. Moreover, TFP growth is more difficult as countries produce more ideas if $\beta > 0$, a term economists call dynamic diminishing returns. Although (7) is simple, it is flexible enough to account for a range of different models of innovation [36] and to estimate trends in productivity growth [28].

We can use estimates of historical factor shares ($\theta_K, \theta_P$) and determinants of future productivity growth ($\lambda, \beta$) to calculate the time it would take for power levels to reach the waste heat limit for a range of future TFP growth rates. First, we solve the TFP growth Eq (7) for $A(t)$. Specifically, we multiply (7) by $A(t)^\beta$ and integrate with respect to time: $A(t) =$

$(\alpha\beta \int_0^t L(s)^\lambda ds + A_0^\beta)^{1/\beta}$, where $A_0$ is TFP at $t_0 = 0$. We choose 2019 to be the initial time ($t = 0$) because that is the last date at which we have an estimate of TFP ($A_0 = 51.5$, S3 Table) from the Penn World Tables (PWT) version 10.01 [33,37]. Substituting in $L(t) = L_0 \exp(g_L t)$, where $L_0$ is $L$ at $t_0 = 0$, yields future TFP as a function of current values of working population and TFP:

$$A(t) = \left( \frac{\alpha\beta L_0^\lambda}{g_N \lambda} (\exp(g_L \lambda t) - 1) + A_0^\beta \right)^{1/\beta} \tag{8}$$

Combining (8) with estimates of $(\lambda, \beta)$, we use (7) to calculate future productivity growth $g_A(t)$.

Second, we solve for power, $P(t)$, given TFP levels, $A(t)$, using (5) and assuming constant growth rates for income, employed population, and capital stock net of power.

$$P(t) = P_0 \exp \left( \frac{1}{\theta_P} [g_Y - \theta_K g_K - \theta_L g_L] t \right) \left( \frac{A(t)}{A_0} \right)^{1/\theta_P}. \tag{9}$$

Third, we use Murphy's [5] waste heat limit from (3).

In order to use (9), we must specify $\lambda$ and $\beta$ values, which are highly uncertain. Bloom et al. [28] estimate $\beta$ by holding the innovative returns to growth in working population ($\lambda$) constant and comparing inputs such as labor and R&D expenditures into innovation and outputs of innovation across sectors and data sets, including outputs such as patents, new drugs, and agricultural land productivity. Their study is among the most comprehensive estimates of deceleration in productivity growth across sectors and time. Given observed R&D input/output ratios, their median estimate across sectors suggests large, self-limiting reductions in research productivity as TFP rises ($\beta = 2.5$) when working population growth provides constant innovative returns ($\lambda = 1$). Bloom et al. [28] cannot identify both $\lambda$ and $\beta$ with data labor inputs and research outputs, so they constrain $\lambda$ and estimate $\beta$. Their baseline value of $\lambda$ is 1, and they obtain estimates of $\beta$ that range from 0.9 to 7.2. We follow their lead and constrain the innovative return to working population at $\lambda = 1$, but consider forecasts under different values of $\beta$ that represent notable or representative economic sectors.

Given the large range in plausible values of $\beta$, Fig 2 explores how different paths for future TFP growth affect the date of the waste heat limit. It shows 6 paths demonstrating that even small changes in $\beta$ have an immense impact on the time to the waste heat limit:

1. A path that continues historical rates of TFP growth: $g_A(t) = 0.01$ (red dashed);
2. A $(\lambda, \beta)$ pair that produces a TFP growth path that approximates historical growth rates: $\lambda = 1, \beta = 1.045$ (blue);
3. The median of the sector-specific $(\lambda, \beta)$ estimated by Bloom et al. [28]: $\lambda=1$, $\beta=2.5$ (green), which corresponds roughly to the path of cotton production according to [28].
4. A path with equal values of $\lambda = 1$ and $\beta = 1$ (purple), which is roughly the path of publicly listed U.S. firms according to [28].
5. Two paths, with $\lambda = 1$ but $\beta$ just below and just above 1 to illustrate how sensitive future power consumption is to this parameter: $\beta = 1.1$ in orange and $\beta = 0.9$ in cyan.

Forecasts start at 2019 levels of working population, power consumption and TFP (see S3 Table). Forecasts assume that production is governed by a Cobb-Douglas function (5) with factor shares estimated using data from 1965–2019 (see S2 Table); that TFP levels are governed by Eq (8), TFP growth by Eq (7), power by Eq (9), and power growth is derived from

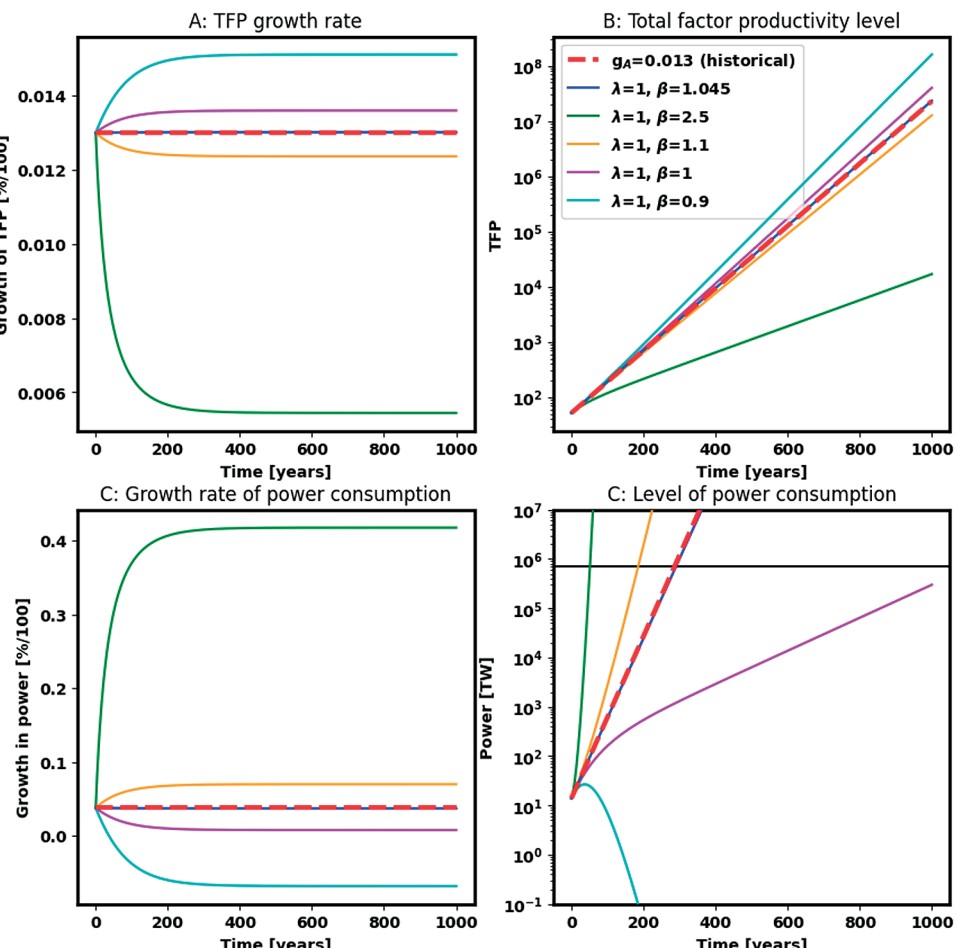

**Fig 2. Projected TFP growth** ($g_A(t)$)**, TFP levels** ($A(t)$)**, growth in power consumption** ($g_P(t)$)**, and power levels** ($P(t)$) **over 1000 years under different parameter values for innovative returns to working population levels** ($\lambda$) **and dynamic diminishing returns to innovation** ($\beta$)**.** The legend provides parameter values for each projection. The black line indicates the waste heat limit. Forecasts start at 2019 levels of working population, power consumption and TFP (see S3 Table). Forecasts assume that production is governed by a Cobb-Douglas function (5) with factor shares estimated using data from 1965–2019 (see S2 Table), that TFP growth is governed by (7), and that output, working population and factor growth rates follow historical values from 1965–2018 (see S2 Table).

the time derivative of Eq (9); and that output, working population and growth rates of factors other than power follow historical values from 1965–2018 (see S2 Table).

Path 1 (red dashed) shows what happens when we use a Cobb-Douglas function and project out using historical estimates of TFP growth. Historical rates of TFP growth portend the waste heat limit in ∼300 years, similar to what Murphy predicts. This forecast implies that a Cobb-Douglas model and a focus on TFP growth can approximate forecasts based on historical growth in average product. The fact that we can approximate historic path 1 with path 2 ($\lambda = 1, \beta = 1.045$, in blue) demonstrates that we can augment the Cobb-Douglas function with the TFP growth model in (7) and match our forecasts to Murphy's [5] forecast.

Path 3 (green) depicts strongly diminishing rates of productivity improvements, implied by median ($\lambda, \beta$) estimates across sectors from Bloom et al. [28]. The waste heat limit occurs in < 100 years in this scenario. Path 4 (purple) illustrates a balanced path in which employed

population growth pushes innovation roughly at the rate that past innovation increases the difficulty of future innovation. This balanced path delays the waste heat limit beyond a millennium. Finally, paths 5 and 6 illustrate how small changes in parameters around a balanced path can have massive effects on forecasts of the waste heat limit. Slightly greater dynamic diminishing returns to innovation ($\beta$ = 1.1) triggers the waste heat limit in under 200 years, while slightly smaller dynamic diminishing to innovation ($\beta$ = 0.9) means historic economic and employed population growth rates never drive power usage to the waste heat limit.

The projected date of the waste heat limit is highly sensitive not just to values of ($\lambda, \beta$), but also to factor shares and economic growth. Fig 3 explores this sensitivity by perturbing these model parameters from their historical values. Panels A and B show that as the factor share of capital or power rises at the expense of labor share, the date of the waste heat limit is delayed. This is because employee income growth is increasing in capital and power factor shares (6), so those shares reduce pressure on power growth to meet the requirement of constant growth in employee income. The black vertical lines and dashed lines in each figure indicate our estimates of the mean and confidence intervals for relevant factor shares. In Panel A, the confidence intervals span a wide range of implied waste heat limit dates. In panel B, the range is quite small. Panels C and D show that increasing the innovative return to working population growth ($\lambda$) increases $\tau$ and diminishing innovative returns with greater innovations ($\beta$) reduces the time to the waste heat limit. The vertical lines in panels C and D show the estimates of $\lambda$ = 1 and $\beta$ = 1.045 that track historical power growth rates. An important aspect of Fig 3 is that the system contains a singularity such that small changes in the parameters can lead to an infinite time to the heat limit, as demonstrated by the path with ($\lambda, \beta$)=(1,0.9) in Fig 2. We interpret these infinite time horizons as indications in our simple model that the waste heat limit is not an important constraint on at least a millennial timescale.

The effect of the growth rate of the working population on the time to the waste heat limit is particularly interesting. Fig 3E shows that, consistent with the intuition of Malthusians [2, 4,5], there is a solution in which significant (employed) population reduction (negative $g_N$) forestalls the waste heat limit. However, consistent with economic intuition [6], there is also a solution in which faster (employed) population growth can forestall the waste heat limit just as effectively. If innovation from more people can outpace the rate at which innovation reduces research productivity, then the waste heat limit can be substantially delayed in the model. But the model in (7) suggests there are two ways for working population-based innovation to keep pace: either increase the innovative returns to working population ($\lambda$, as illustrated in Fig 3C) or increase population growth (Fig 3E). Although total (and thus employed) population growth rates are slowing, the recent historical rate of working population growth (black line) is so close to the elevated level required to evade the waste heat limit that it may be feasible to pursue the optimistic scenario with both population and economic growth.

Of course, one should not view these waste heat limit dates as specific quantitative predictions. They are based on low-order models that are valuable primarily for the qualitative intuition they provide. Our goal is to show, first, that an alternative solution exists in which survival is coupled with perhaps a millennium or more of economic and population growth, and second, that much more research is required before one can make credible predictions about the waste heat limit and the end of economic growth.

## Growth due to substitution to other resources

One might object that inputs into production of power are but one set of limited resources. Even if limits on power turn out not to constrain growth, perhaps other resources could. The

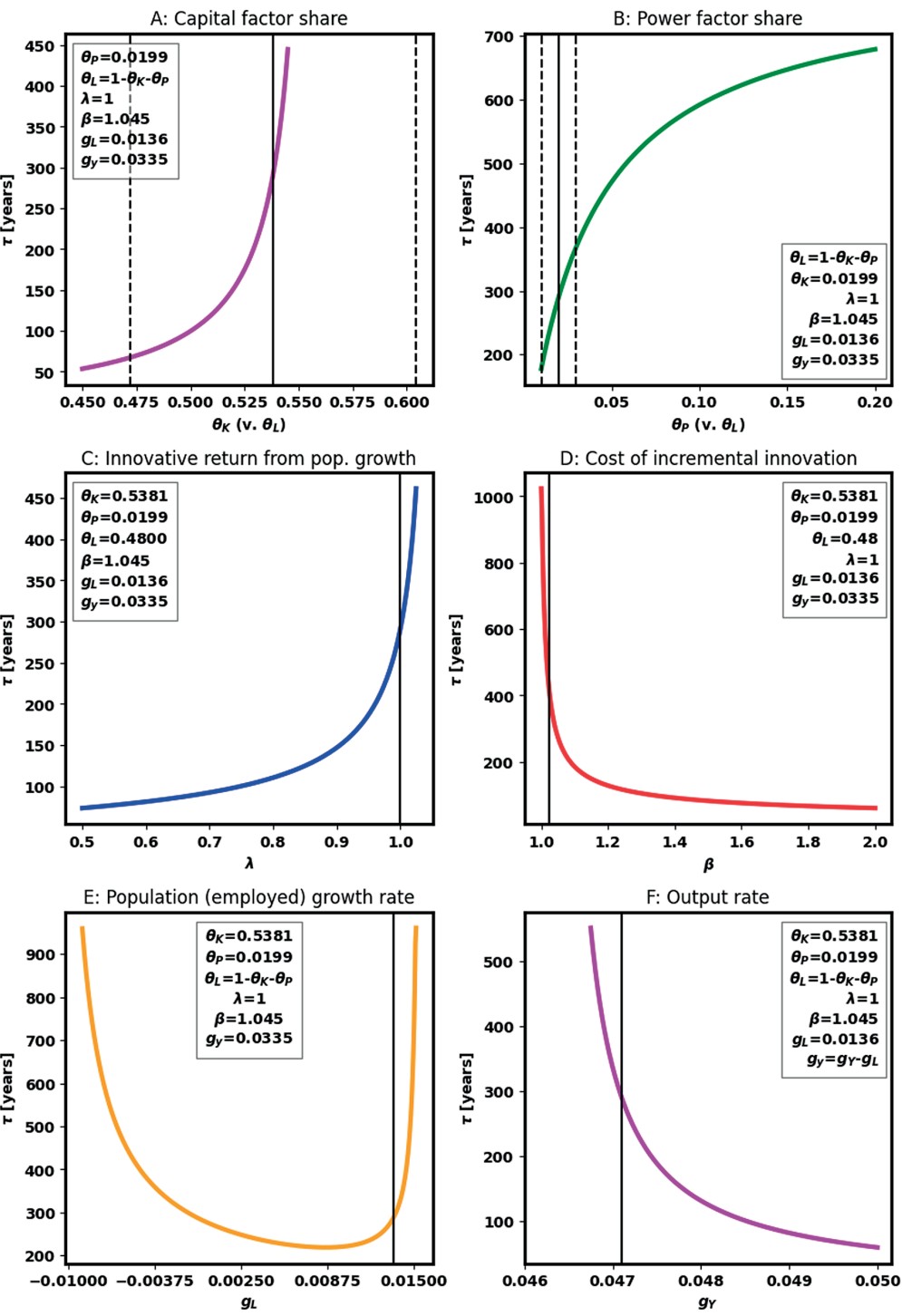

**Fig 3. Projected time until the waste heat limit under different parameter values for capital ($\theta_K$) and power ($\theta_P$) factor shares, innovative returns to working population levels ($\lambda$), dynamic diminishing returns to innovation ($\beta$), growth in working population ($g_L$) and growth in total output ($g_Y$).** Legends provide parameter values for each projection. Solid and dashed black lines in Panels A-D present historical mean and 95% confidence intervals for the parameter specified on the x-axis. The solid black lines in Panels E-F present the historical mean growth in working population and output, respectively. Forecasts start at 2019 levels of working population, power consumption and TFP (see S3 Table). Forecasts assume that production is governed by a Cobb-Douglas function (5) with factor shares estimated using data from 1965–2019 (see S1 Table), that TFP growth is governed by (7), and that output, working population and factor growth rates follow historical values from 1965–2018 (see S2 Table). Each plot changes just the parameter indicated on the x-axis, holding other parameters constant.

converse of this warning is that, even if we run out of specific resources on Earth, we may be able to continue to grow per capita income by substitution to other resources.

In this section, we demonstrate how flexible production technology has historically been, and then discuss how production might adapt to permit economic growth even as certain resources become depleted. We show that historical growth in average productivity combined with stability in resource prices demonstrates a capacity for human civilization to substantially improve how efficiently it uses specific inputs or change how it combines inputs to produce outputs. These adaptations suggest that exhaustion of specific resources, which will raise input prices and perhaps even block improvements in input efficiency, need not stop per capita income growth.

For this section, we adopt a more flexible production function that embeds the Cobb-Douglas function used in the prior section and allows us to see what drives substitution of resource inputs [29]. The constant elasticity of substitution (CES) function is:

$$Y = \left( A\left[ \sum_{i=1}^{R} \theta_i (A_i X_i)^\rho \right] \right)^{1/\rho}, \tag{10}$$

where $i$ indexes a total of $R$ resource inputs $\{X_i\}$, $\{\theta_i\}$ are effective factor shares, $\{A_i\}$ are factor-specific productivities (in contrast to total factor productivity or average product of factors), $\rho = (\sigma - 1)/\sigma$, and $\sigma$ is the elasticity of substitution between resources.

A production process can be thought of as converting a series of inputs (e.g., copper) into a service or function (like conductivity) that helps build an output (such as a cable). Think of $A_i$ as the effective amount of an input $X_i$ required to produce service $i$, i.e., the amount of conductivity extracted from a unit of copper. This could change if one figures out a better way to use copper to create conductivity or finds an alternative to copper that is more conductive and/or more abundant. By contrast, $\theta_i$ is the relative importance of an intermediate service in the production of the output. This could change as cables go from copper to fiber optic for certain purposes such as transporting information. There may be changes in input productivity ($A_i$) or in the conversion of intermediate services as reflected in input shares ($\theta_i$) over time. Going forward we drop the total factor productivity term ($A$) because it can be absorbed into the input-specific productivities ($A_i$).

If $\sigma \geq 1$ (and so $\rho \in (0, 1]$), factors can substitute for each other, with greater $\sigma$ implying more substitutability. As $\sigma \to \infty$, any amount of factor $i$ can be replaced with factor $j$ at some constant rate. If $\sigma < 1$, factors are less easy to substitute: the more one wants to substitute factor $i$ for factor $j$, the more units of $i$ are required for each unit of $j$.

If firms in the economy choose inputs to minimize costs, due to either market pressures or effective internal management, then they will equate the relative marginal product (marginal rate of technical substitution) and the relative prices of resources (marginal rate of substitution): $M_i/M_j = r_i/r_j$ for resource inputs $i$ and $j$, where $M_i = \partial Y/\partial X_i$ indicates marginal product of input $i$, and $r_i$ is the resource price of input $i$. The relative marginal product of inputs $i$ and $j$ with CES production is $\ln M_i/M_j = \ln(\theta_i/\theta_j) + \rho \ln(A_i/A_j) + (1 - \rho)\ln(\overline{A_i}/\overline{A_j})$, where $\overline{A_i} = Y/X_i$ is the average product of resource $i$. If we plug the equation for relative marginal product into the cost minimization condition, take logs, and then take a time derivative, we find that

$$g_{r_{ij}} = g_{\theta_{ij}} + \rho g_{A_{ij}} + (1 - \rho)g_{\overline{A_{ij}}}, \tag{11}$$

i.e., input shares are chosen so that the growth in relative resource price $g_{r_{ij}}$ is equal to growth in relative effective factor shares $g_{\theta_{ij}}$ and the weighted average of growth in relative factor productivity $g_{A_{ij}}$ and in average factor productivity $g_{\overline{A_{ij}}}$.

Recent historical data on resource prices and average factor productivities suggests that the global economy has experienced either a substantial change in effective input mix or improvement in specific input productivities. Input prices have been largely constant for decades, if not centuries [38], implying $g_{r_{ij}} = 0$. Moreover, growth in average productivity has varied substantially across resources (Fig 4). Suppose we examine a resource $i$ that has experienced *slower* average productivity growth relative to input $j$, i.e., $g_{\overline{A_{ij}}} < 0$. Then the ratio $X_j/X_i$ is also declining and (11) implies

$$-g_{\overline{A_{ij}}} = g_{X_{ij}} = \sigma g_{\theta_{ij}} + (\sigma - 1)g_{A_{ij}}, \tag{12}$$

Slower growth in relative average productivity of factor $i$ implies greater use of that input ($-g_{\overline{A_{ij}}} = g_{X_{ij}}$) because the ratio of average productivity of inputs $i$ and $j$ is equal to the ratio of inputs $j$ and $i$ ($\overline{A_i}/\overline{A_j} = (Y/X_i)/(Y/X_j) = X_j/X_i$). Greater relative use of input $i$ can be explained by a mix of substitution towards that input ($g_{\theta_{ij}} > 0$) and improvement in the efficiency with which an input produces that factor ($g_{A_{ij}}$). The mix is governed by the elasticity of substitution ($\sigma$), because $1/(1 - \rho) = \sigma$ and $\rho\sigma = 1 - \sigma$. Assuming inputs are substitutes, lower substitution elasticity implies that changes in factor mix are driven more by change in factor shares than by relative input-specific productivities.

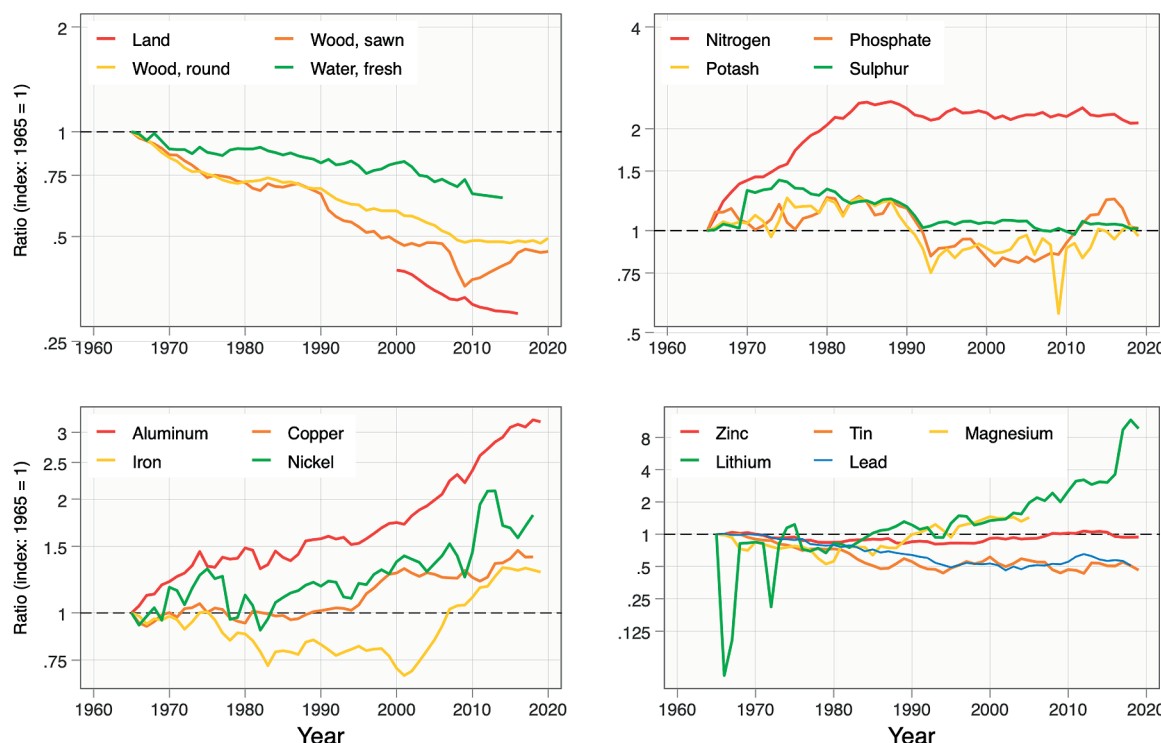

**Fig 4. Use of 18 different commodities, relative to use of power, over time.** Legends indicate specific commodities. Ratio is units of specific commodity divided by units of power, with the ratio normalized to be 1 in 1965. Data are from the US Geological Survey.

Suppose that, in the long-run, production technology is best described by a Cobb-Douglass function ($\sigma = 1$) but with changing shape (i.e., changing factor shares). This assumption is one way to reconcile historical evidence on changing factor shares [30,39]. It is also consistent with a Pareto distribution of ideas that can lead to productivity improvements [32]. With unit elasticity of substitution, past changes in average productivity over time suggest one-for-one changes in factor shares as (12) collapses to $-g_{\overline{A_{ij}}} = g_{X_{ij}} = g_{\theta_{ij}}$. In words, the long-term decrease in relative use of factors such as land and lead and increase in relative use of aluminum and lithium suggest substantial substitution away from the former factors towards the latter.

Indeed, historical changes in input proportions are consistent with factor shares changing quickly. Using factor shares for power as a benchmark, Table 1 suggests that the factor share for land fell on average 1.83% per year and for aluminum rose on average 2.25% per year in the last 60 years. This implies that the factor share of power rose 3x relative to land over this period, while the factor share of aluminum rose nearly 4x relative to power. If we ignore pre-1975 dips in lithium production, use of lithium grew roughly 8x relative to power.

Past is not necessarily prologue, as suggested by the non-monotonicity in resource ratios in Fig 4. However, historical rates of change suggest a degree of plasticity in technology that inspires confidence in humankind's ability to adapt to resource limits without sacrificing economic growth, although this process can involve shocks and disruptions.

## Discussion

Warnings of imminent physical limits to economic growth typically assume exponential population growth coupled with zero or limited growth in productivity and no resource substitution [2,4,5]. This leads to exponential growth in resource usage and dire predictions of

**Table 1. Growth rate in quantity used, average productivity, and factor share ratios (relative to power) for 18 commodities over time.**

| Resource | Time frame | Geometric mean of growth rate of | | |
| --- | --- | --- | --- | --- |
| | | $X_i$: Quantity input used (%/year) | $Y/X_i$: Average product (%/year) | $X_{iP} = \theta_{iP}$: Relative factor shares (%/year) |
| Land | 1960–2016 | 0.07 | 3.85 | −1.83 |
| Woodsawn | 1961–2021 | 0.71 | 3.39 | −1.49 |
| Tin | 1960–2019 | 0.82 | 3.32 | −1.43 |
| Woodround | 1961–2021 | 0.76 | 3.37 | −1.36 |
| Lead | 1960–2018 | 1.02 | 3.15 | −1.28 |
| Waterfresh | 1960–2014 | 1.6 | 2.78 | −0.95 |
| Zinc | 1960–2019 | 2.42 | 1.7 | −0.13 |
| Potash | 1960–2019 | 2.6 | 1.52 | −0.03 |
| Power | 1965–2021 | 2.07 | 1.89 | 0 |
| Sulfur | 1960–2019 | 2.41 | 1.71 | 0.04 |
| Phosphate | 1960–2019 | 2.91 | 1.22 | 0.04 |
| Lithium | 1960–2019 | 5.52 | −1.28 | 0.4 |
| Iron | 1960–2019 | 2.68 | 1.44 | 0.48 |
| Copper | 1960–2018 | 2.88 | 1.29 | 0.65 |
| Magnesium | 1961–2005 | 3.61 | 0.75 | 0.84 |
| Nickel | 1960–2018 | 3.54 | 0.64 | 1.17 |
| Nitrogen | 1960–2019 | 4.15 | 0.01 | 1.42 |
| Aluminum | 1960–2019 | 4.58 | −0.4 | 2.25 |

Notes. The factor share ratio for a commodity is equal to the factor share for that commodity divided by the factor share for power. Factor share ratios are equal to input ratios. The input ratio for a commodity is the ratio of the quantity of that commodity used in a year and the quantity of power used that year. Data are from the US Geological Survey.

impending physical limits to economic growth and perhaps the extinction of humans. We acknowledge that there are large uncertainties in future improvements to productivity and resource substitution, but show that reasonable projections based on historical data can delay the predicted date of physical limits to economic growth for a thousand years, even if humans remain bound to Earth. To put it another way, *The Limits to Growth* literature predicts a short period of future economic growth with small uncertainty, but we argue that if increases in productivity and resource substitution are properly accounted for, the potential period of future economic growth reaches at least the millennial timescale, although it also becomes more uncertain.

Murphy [5] also argues that increases in efficiency are theoretically bounded, giving physical examples such as the amount of light produced per Watt of power. But the utility flow from goods and services does not depend simply on energy usage. For example, when business is conducted and experiences are provided virtually, GDP can be increased even as energy usage is decreased. Moreover, physical goods, such as molecules, can be rearranged in ways that dramatically increase their value. For example, when a new pill is invented that substantially reduces pain, disability, or dementia in old age, this adds immense GDP at trivial incremental energy usage [40,41]. So Murphy's objection results from the category error of applying physical arguments to GDP, which is not a physical quantity.

The functional form assumptions we make when conducting our simulation exercises do not qualitatively undermine our conclusion about the uncertainty about when limits to growth will bind. For example, our analysis employs production functions – Cobb-Douglas (CD) or constant-elasticity-of-substitution (CES) – that Baumgärtner [42], Meran [43] and others have shown do not obey the laws of thermodynamics. However, use of the alternative production functions, including CES with a thermodynamic constraint, recommended [43] as substitutes would yield qualitatively similar results, i.e., substantial sensitivity to critical parameter values. Moreover, we neither assume nor conclude that there are no limits to growth. Rather, our view is that the existence of asymptotic limits on economic models does not imply that these constraints are imminent or require immediate curbs on economic activity. We show that it is possible (but not certain) that physical limits to economic growth are many hundreds or thousands of years in the future.

Because our model is primary physical, rather than economic, our conclusions about time limits may be conservative. For example, none of the arguments we present account for the price of resources, which could further reduce the risk of exhaustion and encourage substitution. This point was inspired by Hotelling [44] and made explicit by Simon [6]. One shortcoming of price rationing is that some limits are not priced. For example, Murphy highlights temperature change induced by power production, a social cost not automatically included in the price of inputs into power or of power. For prices to control temperature change due to power use, a society or government would need to impose, for example, Pigouvian taxes [45] to account for the negative externality of the temperature cost of power. Broader measures of a society's productivity could include these if output measures included external effects such as climate change, but they currently do not.

Although our analysis suggests that physical limits to growth are both highly sensitive to critical parameter values and potentially far off in the future, there remains merit to the limits to growth argument for some variables on long enough timescales. For example, at a population growth rate of 1%/yr the mass of humans would equal the mass of Earth in approximately 3000 years and the mass of the observable universe in approximately 9000 years. Moreover, given that a human has a power output of $\sim$100 W, the power output of humanity would hit the waste heat limit in 1400 years at a population growth rate of 1%/yr. So population is physically limited in a way that per capita income is not. That said, this

point should not be taken as requiring immediate restrictions on economic and population growth.

The appropriate policy response in the face of the wide array of possible futures we present is an open question. A naive application of the precautionary principle [46] might suggest strict controls on power and economic growth in the face of any risk of dire consequences. But for individuals not yet born or individuals driven to poverty and death [47], precautionary limits on power usage or growth also pose existential risk. Moreover, resource depletion is not the only existential risk. Other risks include pandemics, artificial general intelligence, asteroid impact, and colonization by aliens [48]. Because economic growth provides the budget to address these other risks [49], the precautionary principle may be indeterminate, or even point in the other direction [50]. Therefore, it is also important to consider alternatives to the precautionary principle, including risk assessment (which takes greater effort to quantify risks) [51], risk-risk assessment (which weighs competing risks) [52], cost-benefit analysis (which considers both the costs and benefits of precautionary or adaptive actions) [49], and welfare frameworks that account for the value of population growth [47,53].

## Supporting information

### The value of a dollar

Dollars are a unit of exchange that, among other things, simplifies the expression of prices. Suppose there are only 2 goods, and the exchange rate is 4 units of good 1 for 3 units of good 2. We can say the relative price of good 2 is 4/3 of good 1: 1 unit of good 2 costs 4/3 units of good 1. If there is also a good 3 that can be exchanged for 2 units of good 1, then the price of good 2 can also be expressed as 1/2 of good 3. However, we can define a medium of exchange like a dollar to simplify prices, which can now all be expressed in dollars rather than in each of the other goods. Suppose we define a dollar to be equal to the value of 1 unit of good 1. Then the price of good 2 is simply $(4/3)≈$1.33 and the price of good 3 is $(2/1)*(4/3)≈$2.67. Because the dollar is no longer pegged to a commodity like gold, the so-called quantity theory of money says that the value of a dollar will be a function of the supply and demand for a dollar as a medium of exchange rather than gold. Although there are other theories of the value of money, they do not affect relative prices, so the analysis in this paper does not depend materially on the theory used.

There are two complications when valuing a dollar. First, the relative dollar prices of specific goods and services depend on the relative demand and supply for those different items. If the quantity of dollars or the quantity of goods and services changes over time, the dollar price of a given item will change. Prices may go up (inflation) or they may go down (deflation). Real prices, defined as prices expressed in the value of a dollar in a given year, address this problem by eliminating temporal changes in the value of a dollar. Second, different locations may use different units of exchange, e.g., the US uses a dollar and England uses a pound. This creates two difficulties: we need to know the amount of dollars one may trade for one pound and relative prices may differ across countries because transportation costs make it difficult for markets to equalize relative prices worldwide. A purchasing power parity (PPP) exchange rate addresses these problems. It is defined as the exchange rate that adjusts for both the nominal currency exchange rate and differences in relative prices for a fixed bundle of commonly consumed goods. Thus, a PPP-adjusted dollar value of output in a country like England in a year is the value of England's output in pounds that year times the PPP exchange rate for dollars to pounds.

**Output, income and welfare.** Economists often use output as a proxy for income, and often even for welfare. The intuition for using output as a measure of income is that

consumers can only consume what is produced. There are typically two criticisms of these equivalences. One is a technical one: output in one period may be consumed in a later period (think inventories) or may produce services that are consumed over time (think durable goods). Inventories are a second order concern because they tend to be a small fraction of consumption, especially over a longer time period, e.g., 5 years. The problem of durable goods is addressed by accounting: capital assets are depreciated (i.e., consumed or reduced in value) at some rate.

The second criticism is that the gross domestic product does not count many goods and services that are not traded. For example, childcare by parents is not included in GDP because its quantity is not measured nor is there a market for pricing a mother's care of her child (as opposed to a babysitter's care for a child). Moreover, many things contribute to welfare that are not a good or a service in the traditional sense. For example, leisure and public safety affect welfare but are not included in output. Economists are aware of these errors and work to mitigate them. For example, economists understand that the market implicitly assigns a value for leisure (wages) and for public safety (capitalized into land prices) and use them to value those items [54].

**Estimates used in the text.** S1 to S3 Tables have estimates used in the text.

**S1 Table. Estimating factor shares from historical, country-year level data from 1965–2019.** Notes. Table provides estimates of historical factor shares, estimated using Eq (7). Specifically, it presents results of a regression of the growth rate of GDP per employee on the growth rate of non-power physical capital per employee and on the growth rate of power per employee. The coefficient on the former regressor yields an estimate of $\theta_K$ and the latter an estimate of $\theta_P$. The constant provides an estimate of the average growth rate of TFP ($g_A$). Data are from Penn World Tables (PWT) 10 and Our World in Data (OWID). The sample includes countries with both (a) income, employee population, and capital stock and (b) power consumption for the entire period from 1965–2019. Data are available for 66 countries from 1965–1991; in 1992 USSR is replaced with 14 countries that replaced the USSR. Moldova is excluded due to lack of data.
(PDF)

**S2 Table. Average growth rates of key parameters for 92 countries form 2000–2018.** Notes. Table provides geometric means of growth rates of per capita income, population, capital stock, and power consumption for 92 countries from 2000-2018 using PWT data. These means are calculated at the country x year level, where countries are weighted by their GDP. 2019 is omitted because we do not have data for 2020 and thus growth rates for 2019. Units are fraction or %/100. These means are used, along with Eqs (8) and (9), to forecast when the waste heat limit will occur in Figs 2 and 3.
(PDF)

**S3 Table. Initial values from 2019 for future projections.** Notes. Table presents the 2019 values of total employment, total population, total primary power consumption, and average TFP for 222 countries. We use these as initial values for the plots in Fig 2. Total employees, population, and power consumption is obtained for all 222 countries for which these data are available in 2019 from PWT 10.0 and Our World in Data. We obtain average TFP in 2 steps. First, we take estimates of factor shares from S1 Table and use them to estimate TFP using Eq (5) for each of the 222 countries in 2019. Second, we take an arithmetic average of those TFPs.
(PDF)

## Acknowledgments

We acknowledge helpful feedback from John Cochrane, RJ Graham, Edwin Kite, Jacob Haqq-Misra, Predrag Popovic and participants at the University of Chicago Law School Work-in-Progress workshop on an early draft of this manuscript. We thank Liam Starnes for research assistance.

## Author contributions

**Conceptualization:** Dorian S. Abbot, Anup Malani.

**Data curation:** Anup Malani.

**Formal analysis:** Dorian S. Abbot, Anup Malani.

**Investigation:** Dorian S. Abbot, Anup Malani.

**Methodology:** Dorian S. Abbot, Anup Malani.

**Software:** Dorian S. Abbot, Anup Malani.

**Visualization:** Dorian S. Abbot, Anup Malani.

**Writing – original draft:** Dorian S. Abbot, Anup Malani.

**Writing – review & editing:** Dorian S. Abbot, Anup Malani.

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
