## [Decision Letter · Decision Letter 0]

1 Jul 2024

PONE-D-24-08954

Revisiting the physical limits to economic growth

PLOS ONE

Dear Dr. Abbot,

Thank you for submitting your manuscript to PLOS ONE. After careful consideration, we feel that it has merit but does not fully meet PLOS ONE’s publication criteria as it currently stands. Therefore, we invite you to submit a revised version of the manuscript that addresses the points raised during the review process.

Dear Dr. Dorian

I hope this message finds you well.

After a thorough review process, I have received feedback from two reviewers regarding your manuscript titled “Revisiting the physical limits to economic growth”. Based on their comprehensive comments and suggestions, I believe that your manuscript holds significant potential and is worth pursuing further. However, substantial revisions are required before it can be considered for publication. Therefore, I'm issuing a "Major Revise and Resubmit" decision. Please carefully address the reviewers' comments and make the necessary revisions to improve the manuscript.

I look forward to receiving your revised submission.

We look forward to receiving your revised manuscript.

Kind regards,

Zakaria Boulanouar, PhD

Academic Editor

PLOS ONE

Journal Requirements:

2. Please include a caption for figure 4.

3. We notice that your supplementary tables are included in the manuscript file. Please remove them and upload them with the file type 'Supporting Information'. Please ensure that each Supporting Information file has a legend listed in the manuscript after the references list.

**Additional Editor Comments:**

Dear Dr. Dorian

I hope this message finds you well.

After a thorough review process, I have received feedback from two reviewers regarding your manuscript titled “Revisiting the physical limits to economic growth”. Based on their comprehensive comments and suggestions, I believe that your manuscript holds significant potential and is worth pursuing further. However, substantial revisions are required before it can be considered for publication. Therefore, I'm issuing a "Major Revise and Resubmit" decision. Please carefully address the reviewers' comments and make the necessary revisions to improve the manuscript.

I look forward to receiving your revised submission.

Regards

Zakaria

Reviewers' comments:

Reviewer's Responses to Questions

**Comments to the Author**

1. Is the manuscript technically sound, and do the data support the conclusions?

Reviewer #1: Yes

Reviewer #2: Partly

2. Has the statistical analysis been performed appropriately and rigorously? 

Reviewer #1: Yes

Reviewer #2: Yes

3. Have the authors made all data underlying the findings in their manuscript fully available?

Reviewer #1: Yes

Reviewer #2: Yes

4. Is the manuscript presented in an intelligible fashion and written in standard English?

Reviewer #1: Yes

Reviewer #2: Yes

5. Review Comments to the Author

Reviewer #1: The purpose of this manuscript is to evaluate the validity of the hypothesis proposed by Murphy (2022: Nature Physics) that waste heat from electricity consumption may determine the physical limit to economic growth. The authors used simple economic and physical models to represent the relationship between economic growth and energy consumption (i.e., waste heat), and calculated a time until the Earth's surface becomes uninhabitable due to high temperatures. As a result, the authors found that waste heat from electricity consumption will not necessarily limit economic growth because population growth can increase total factor productivity (TFP). Since the economic model used in this study is quite simple, it would be difficult to derive policy implications from the calculation results. However, this manuscript provides new insights into Murphy's hypothesis. I have several questions about the methodology. Comments are provided below.

Comment #1: This study assessed the impact of waste heat from electricity consumption on the Earth's surface temperature. I could not understand why waste heat from direct combustion of fossil fuels was not considered. Please explain the reason.

Comment #2 (page 5, lines 141-161): The authors analyzed the relationships between electricity consumption, population, per capita income, and energy efficiency using time series data from 1965 to 2020. By comparing the average growth rates of the socioeconomic variables, the authors found that population and economic growth were possible without growth in electricity consumption. I am concerned that this result depends on the choice of data period. Is the authors' finding robust to changes in the data period?

Comment #3 (page 5, Figure 1): The plot of gP, gN, gy, and gAp may be more appropriate for the context of the manuscript than the plot of P, N, y, and Ap.

Comment #4 (page 5, lines 141-161): The manuscript says: "Evidently, current increases in efficiency are capable of driving significant economic and population growth even in the absence of growth in power consumption." This statement seems odd. Equation (2) is an identity and always holds mathematically. However, it does not mean that we can freely choose a combination (gN, gy, gAp) that satisfies 0 = gN + gy - gAp (How can we control the growth rates?). The above statement is mathematically true, but may not be feasible in the real world.

Comment #5 (pages 7-8, lines 271-291): The authors estimated the parameters of the production function (Equation (5)) using panel data for many countries. However, the differences in the parameters across countries are not examined in this study. Is there any reason why the authors used country data for parameter estimation instead of aggregated global data?

Comment #6 (page 8): The authors should show that the TFP growth model expressed in Equation (7) is able to explain historical changes in TFP. Please estimate the parameters of the TFP growth model using historical data and summarize the result in the SI.

Reviewer #2: For the authors, the aim of their article is to propose a common framework for resolving the disagreement on the question of possible limits to growth. For the reasons that I explain below, I don't think at all that this article achieves this objective.

First of all, the authors pit natural scientists against economists, whereas there is also disagreement between economists on this issue. Secondly, they caricature the debate between the two camps, suggesting that the first camp ignores productivity improvements and substitution possibilities (between natural and man-made inputs). In the contemporary view, the proponents of possible limits to growth do not deny either of these possibilities but put forward arguments, based on the laws of physics, to explain why these two types of possibilities are limited. The authors never refer to these arguments. Thereby, the proposed framework seems hardly adequate to resolve the disagreement it seeks to resolve.

Moreover, for the main part of their paper, 1) the authors make the choice of a technological assumption (Cobb Douglas technology), which is known to be inconsistent with the implications of the law of physics (see e.g. Baumgärtner (2004), Energy and Resource Economics, Meran (2019), Energy Economics); 2) they assume that technical progress is a process that only requires labour (see equation (7)) and does not consume any resource (here power).

The combination of such types of assumptions is known to be likely to produce more favourable results to growth since assumption (1) amounts to assuming that the marginal productivity of resources is unbounded and assumption (2) implies that there is no limit to technological progress. By this same token, the article unsurprisingly leads to rather comforting results about our growth prospects.

It is however fair to say that some results of the sensitivity analysis proposed by the authors are less reassuring: when beta is below 1, i.e. when technical progress is less easy, things get more complicated. This is precisely one of the 2 points I raised above...

For the main part of their paper, the authors are only interested in one type of resource: "power". No distinction is e.g. made between energy and non energy resource. They do not motivate why restricting theirselves to a single dimension is enough to offer a right framework for the ambitious goal they announce.

The status of the section entitled ``Growth due to substitution to other resources'' from lines 406 to 493 is unclear to me even though it implicitly aims at answering to my previous point of criticism. But methodologically, this section is not really linked to the approach of the previous ones. By using a multi-input technology, they want to broaden the range of resources they take into account, whereas until now they have only talked about power. It is a sort of appendix in which the authors confirm that in the past there have been substantial changes in the input mix used. This shows that there has been a good "degree of plasticity in technology". I have no difficulty with that claim based on past evidence. But after this observation, the section suddenly ends with a profession of faith: ``this technological plasticity inspires confidence in humanking's ability to adapt to resource limits without sacrificing economic growth...''. This statement is a conviction and not something that the section demonstrates. This section does not even explicitly address the question of resource limits.

More incidental comments

In the current state of affairs, the paragraph on lines 25 to 47 is a purely speculative and useless detour. I do not see the point of alluding to it. In any case, it does not contribute to ``mediating the dispute" that is the main focus of the paper.

Lines 80 and 81: what does the acronym SI mean?

Line 188: Murphy's assumption is made for the sake of illustration as a "what if" exercise. Contrary to what the auhors seem to suggest, it is not an assumption that he considers to be true (in his article, see the comment in the second paragraph below equation 2).

Line 136: the authors state that the units of variable $W$ are

The intuitive reasoning leading to equation (4) could be made more explicit.

Computations underlying Figures 2 and 3: the authors could be more explicit about the way they compute $g_Y$ and $g_K$.

Line 387 about the U-shaped curve on panel F of figure 3: the authors consider this shape as particularly interesting. I think that the right-hand part of the U-shape is a direct consequence of the assumption they make on the law of motion of technological progress.

Supporting information from lines 553 to 603. These two sections (in particular the second one) are really elementary. I don' t them think they are very useful here.

6. PLOS authors have the option to publish the peer review history of their article (what does this mean?). If published, this will include your full peer review and any attached files.

Reviewer #1: No

Reviewer #2: No

---

## [Decision Letter · Decision Letter 1]

7 Oct 2024

PONE-D-24-08954R1

Revisiting the physical limits to economic growth

PLOS ONE

Dear Dr. Abbot,

Thank you for submitting your manuscript to PLOS ONE. After careful consideration, we feel that it has merit but does not fully meet PLOS ONE’s publication criteria as it currently stands. Therefore, we invite you to submit a revised version of the manuscript that addresses the points raised during the review process.

We look forward to receiving your revised manuscript.

Kind regards,

Zakaria Boulanouar, PhD

Academic Editor

PLOS ONE

Additional Editor Comments:

Dear Professors Abbot and Malani,

I hope this message finds you well.

I would like to thank you for your efforts in revising your manuscript, which you submitted for consideration in PLOS ONE.

Reviewer 1 has accepted the revised paper, while Reviewer 2 has provided constructive feedback that necessitates further revisions. Below, I summarize his comments.

First of all, he noted that you have done a "creditable job of revision, although some aspects remain minimalist." Furthermore, he highlighted two main dimensions for assessment: the technical developments and the ambition of proposing "a common framework for resolving the disagreement" surrounding the physical limits to economic growth. Reviewer 2 feels that the article falls short of its intentions in this regard and suggests that a significant amount of rewriting is needed to align the paper more closely with the objectives you have outlined.

He also expressed some additional and more specific comments, which include the following:

1. He mentioned that you have not addressed the optimistic assumption that the innovation process is not resource-consuming, particularly regarding energy usage. This assumption has implications that should be considered in your analysis.

2. The section on growth due to resource substitution remains largely unchanged, and the reviewer’s initial concerns about this section persist.

3. The title should reflect the perspective regarding the physical limits to growth, particularly incorporating the concept of "waste heat limit," as it is central to your findings.

4. Regarding literature references, the reviewer believes that while you reference the 1972 report "The Limits to Growth," he suggests including more recent contributions to the literature, particularly in mathematical ecological economics, to strengthen your arguments.

5. There are contradictory statements in the abstract that should be reconciled to avoid confusion for readers.

6. The reviewer expressed concern about the asymmetry in how economic and physical concepts are presented, questioning the assumption that economists have a better understanding of physics than natural scientists have of economics.

7. Finally, the reviewer pointed out that the assertion regarding GDP growth and reduced energy usage needs further substantiation, especially from a global perspective.

Given the constructive nature of the reviewer’s comments, I invite you to revise your manuscript accordingly and resubmit it for further consideration.

Please let me know if you have any questions or need further clarification; I will be happy to continue acting as the intermediary for the benefit of all stakeholders!

I look forward to receiving your revised manuscript.

Regards,

Zakaria

Reviewers' comments:

Reviewer's Responses to Questions

**Comments to the Author**

1. If the authors have adequately addressed your comments raised in a previous round of review and you feel that this manuscript is now acceptable for publication, you may indicate that here to bypass the “Comments to the Author” section, enter your conflict of interest statement in the “Confidential to Editor” section, and submit your "Accept" recommendation.

Reviewer #1: All comments have been addressed

Reviewer #2: (No Response)

2. Is the manuscript technically sound, and do the data support the conclusions?

Reviewer #1: (No Response)

Reviewer #2: Partly

3. Has the statistical analysis been performed appropriately and rigorously? 

Reviewer #1: (No Response)

Reviewer #2: Yes

4. Have the authors made all data underlying the findings in their manuscript fully available?

Reviewer #1: (No Response)

Reviewer #2: Yes

5. Is the manuscript presented in an intelligible fashion and written in standard English?

Reviewer #1: (No Response)

Reviewer #2: Yes

6. Review Comments to the Author

Reviewer #1: (No Response)

Reviewer #2: The authors have done a creditable job of revision, even if it remains minimalist in some respects (*).

There are at least two dimensions with respect to which this article can be assessed: the technical developments that the authors have carried out and its ability to achieve the ambitious goal of proposing "a common framework for resolving the disagreement" surrounding the question of the physical limits to economic growth. It is on the second level that, in my view, the article remains well below its intentions. On this point, a certain amount of rewriting remains to be done to place the article in a perspective more faithful to the exercise actually proposed by the authors (**).

(*) two main remarks on this point

a) If I am not mistaken, the authors have not made any comment on my remark about their (optimistic) assumption that the innovation process is not resource consuming (here power consuming). It is not incidental: there are various contributions to the economic growth literature showing that global productivity gains could be slowed down by stronger environmental constraints, something which is absent from the innovation process described by equation (7).

b) The section entitled ‘Growth due to substitution to other resources’ remains essentially unchanged. Therefore, quite logically, my point of view about this section and the status of its conclusion remains essentially the same as in my first report.

(**) Here are three suggestions that, in my view, might help

1) The title of the article should make explicit the perspective according to which the physical limits to growth are revisited. The concept of "waste heat limit" should therefore appear in the title because it is in reference to this concept that the [main] results of the paper are obtained.

2) If the authors quote the 1972 report (The Limits to Growth) several times, they cite relatively few later contributions to the literature on environmental limits to economic growth. In particular, they make almost no reference to contributions that could be classified as ``mathematical ecological economics''. Beyond Murphy, they now mention Meran and Baumgärtner, whom I had mentioned in my frist report, but no one else. Even if this field is not dominant in Economics, it is at the very heart of the dispute that the authors want to contribute to resolve. I think it would therefore be justified to make more reference to it. Alternatively, it would be better not to present the article as an attempt at "mediating a dispute" but to focus on the goal of revisiting Murphy's analysis.

3) The authors should avoid statements that could seem to be contradictory, even to a well-meaning reader. In the abstract for instance, the authors outline a "remarkable parameter sensitivity: across a plausible range of parameters waste heat never constrains economic growth or does so in a few hundred years". But a few lines below, they end the abstract by writing "our work highlights the potential of an optimistic future for humanity in which economic growth continues on a millennial perspective". From the former careful sentence to the latter assertive one, it seems to be a big gap. At the very least, the authors should state the conditions under which their last assertion could be met.

Other comments:

As far as the "Supporting information" section is concerned, the authors explain in their letter why they wish to keep this section. If I follow their point, I must conclude that there is then a strange asymmetry in the paper: basic economic concepts are explained in detail while (not so basic) concepts of physics are not. The authors seem to assume that economists know more about Physics than natural scientists know about Economics. I am not as sure as they are...

Line 525: the authors write: "When business is conducted and experiences are provided virtually, GDP can be increased even as energy usage is decreased". Virtually maybe but if I take a global (Worldwilde) perspective, this has never been the case (at least until 2020). At the level of some countries, an absolute decoupling has recently been observed. But it is not the case at the World level (which is, to my eyes, the relevant one for the issue at stake here): until now (or at least until 2020 since I have not checked since then), World economic growth has always gone hand in hand with higher energy consumption (see e.g. bp energy outlook).

7. PLOS authors have the option to publish the peer review history of their article (what does this mean?). If published, this will include your full peer review and any attached files.

Reviewer #1: No

Reviewer #2: No

---

## [Decision Letter · Decision Letter 2]

22 Jan 2025

PONE-D-24-08954R2

Revisiting the physical limits to economic growth

PLOS ONE

Dear Dr. Abbot,

Thank you for submitting your manuscript to PLOS ONE. After careful consideration, we feel that it has merit but does not fully meet PLOS ONE’s publication criteria as it currently stands. Therefore, we invite you to submit a revised version of the manuscript that addresses the points raised during the review process.

**Dear authors**

Thank you again for submitting the revised version of your manuscript.

Following the second round of review, Reviewer 2 has provided additional feedback. Specifically, the reviewer suggests that the title of the manuscript should better reflect the specific focus of your work, namely revisiting the question of the limits to growth through the lens of the waste heat limit. This adjustment would ensure clarity and accuracy in conveying the scope of the research to readers.

While the reviewer acknowledges that you have provided explanations for why certain comments were not addressed, they have also indicated that their primary concern is with the alignment of the title with the content. Addressing this point will likely enhance the manuscript’s impact and its reception by the readership.

To proceed, please revise the manuscript title to more explicitly reflect its focus on the waste heat limit as a prism for examining the limits to growth. Also, if necessary, make minor adjustments in the text to ensure the alignment of the title and content.

I look forward to receiving your revised manuscript.

Regards. 

We look forward to receiving your revised manuscript.

Kind regards,

Zakaria Boulanouar, PhD

Academic Editor

PLOS ONE

**Journal Requirements:**

**Additional Editor Comments:**

Dear authors

Thank you again for submitting the revised version of your manuscript.

Following the second round of review, the Reviewer  has provided additional feedback. Specifically, the reviewer suggests that the title of the manuscript should better reflect the specific focus of your work, namely revisiting the question of the limits to growth through the lens of the waste heat limit. This adjustment would ensure clarity and accuracy in conveying the scope of the research to readers.

While the reviewer acknowledges that you have provided explanations for why certain comments were not addressed, they have also indicated that their primary concern is with the alignment of the title with the content. Addressing this point will likely enhance the manuscript’s impact and its reception by the readership.

To proceed, please revise the manuscript title to more explicitly reflect its focus on the waste heat limit as a prism for examining the limits to growth. Also, if necessary, make minor adjustments in the text to ensure the alignment of the title and content.

I look forward to receiving your revised manuscript.

Regards

Reviewers' comments:

Reviewer's Responses to Questions

**Comments to the Author**

1. If the authors have adequately addressed your comments raised in a previous round of review and you feel that this manuscript is now acceptable for publication, you may indicate that here to bypass the “Comments to the Author” section, enter your conflict of interest statement in the “Confidential to Editor” section, and submit your "Accept" recommendation.

Reviewer #2: (No Response)

2. Is the manuscript technically sound, and do the data support the conclusions?

Reviewer #2: Partly

3. Has the statistical analysis been performed appropriately and rigorously? 

Reviewer #2: Yes

4. Have the authors made all data underlying the findings in their manuscript fully available?

Reviewer #2: Yes

5. Is the manuscript presented in an intelligible fashion and written in standard English?

Reviewer #2: Yes

6. Review Comments to the Author

**Reviewer #2: **(No Response)

7. PLOS authors have the option to publish the peer review history of their article (what does this mean?). If published, this will include your full peer review and any attached files.

Reviewer #2: No

---

## [Author Response · Author response to Decision Letter 3]

25 Jan 2025

We have changed the title of our paper to “Revisiting the physical limits to economic growth, with a focus on the waste heat limit.”

---

## [Editor Report · Decision Letter 3]

29 Jan 2025

Revisiting the physical limits to economic growth, with a focus on the waste heat limit

PONE-D-24-08954R3

Dear Dr. Abbot,

We’re pleased to inform you that your manuscript has been judged scientifically suitable for publication and will be formally accepted for publication once it meets all outstanding technical requirements.

Kind regards,

Zakaria Boulanouar, PhD

Academic Editor

PLOS ONE

Additional Editor Comments (optional):

Thank you for revising the title of your manuscript.
---

## [Editor Report · Acceptance letter]

PONE-D-24-08954R3

PLOS ONE

Dear Dr. Abbot,

I'm pleased to inform you that your manuscript has been deemed suitable for publication in PLOS ONE. Congratulations! Your manuscript is now being handed over to our production team.

Kind regards,

on behalf of

Dr. Zakaria Boulanouar

Academic Editor

PLOS ONE